# Inequalities in the Distribution of the Nursing Workforce in Albania: A Regional Analysis Using the Gini Coefficient

**DOI:** 10.3390/nursrep15020030

**Published:** 2025-01-22

**Authors:** Blerina Duka, Alketa Dervishi, Eriola Grosha, Dhurata Ivziku, Gennaro Rocco, Alessandro Stievano, Ippolito Notarnicola

**Affiliations:** 1Faculty of Medicine, Catholic University “Our Lady of the Good Counsel”, 1000 Tirana, Albania; bleriduka@yahoo.it; 2Faculty of Technical Medical Sciences, University of Medicine, 1005 Tirana, Albania; alketa.dervishi@umed.edu.al; 3University of Rome “Tor Vergata”, 00133 Rome, Italy; e.grosha8400@stud.unizkm.al; 4Department of Health Professions, Fondazione Policlinico Universitario Campus Bio-Medico, 00128 Rome, Italy; d.ivziku@policlinicocampus.it; 5Centre of Excellence for Nursing Scholarship, OPI, 00136 Rome, Italy; gennaro.rocco@uniroma2.it (G.R.); alessandro.stievano@unime.it (A.S.); 6Department of Clinical and Experimental Medicine, University of Messina, 98122 Messina, Italy; 7Department Medicine and Surgical, University of Enna “Kore”, 94100 Enna, Italy

**Keywords:** nursing workforce, health disparities, regional inequalities, human development index, Gini coefficient, rural healthcare, healthcare access, nurse distribution, healthcare management

## Abstract

**Background/Objectives:** The uneven distribution of nurses in Albania is a major problem that compromises equitable access to health services. Rural and less developed regions suffer from a chronic shortage of nursing staff, while urban areas attract health professionals. This study aims to quantify the inequalities in the distribution of nurses in Albania, analyzing the nurse-to-population ratio and its impact on the quality of healthcare. The main objective of this study is to examine the distribution of the nursing workforce in Albania and assess regional disparities, using the Gini coefficient and the Human Development Index (HDI) to measure and compare inequalities between regions. **Methods:** This descriptive–analytical study was conducted in 2024. The data were collected from official sources, including the Albanian Ministry of Health and the World Health Organization (WHO). The Gini coefficient and the Lorenz curve were used to analyze the distribution of nurses in relation to the population and HDI of the different regions. The analysis included data on the number of nurses, population, and regional socioeconomic conditions. **Results:** The average nurse-to-population ratio in Albania is 28 nurses per 10,000 inhabitants, with significant variations between regions. Tirana has the highest ratio (60 nurses per 10,000 inhabitants), while Kukës and Dibër have the lowest values (10 per 10,000 inhabitants). The calculated Gini coefficient is 0.0228, indicating a very low level of inequality in the distribution of the nursing workforce. **Conclusions:** Inequalities in the distribution of nurses in Albania require targeted policy interventions. Policies are needed that incentivize health workers to work in less developed regions, through economic incentives, infrastructure improvements, and lifelong learning programs. These interventions are essential to reduce disparities and ensure equitable access to health services across the country.

## 1. Introduction

Uneven access to human resources in the health sector is a complex challenge in many countries, especially developing countries, and Albania is no exception [1]. The uneven distribution of nursing staff can drastically reduce access to care services and compromise the overall quality of healthcare services offered to the population [2]. Nurses constitute an essential component of the healthcare system, playing a crucial role not only in direct patient care, but also in disease prevention, treatment management, and rehabilitation programs [3,4]. Their presence, distribution, and expertise are therefore crucial for the effectiveness of the health system and for ensuring equitable access to quality health services [5]. However, when the nursing workforce is not equally distributed, inequalities are generated that penalize particularly vulnerable areas, with serious consequences for the well-being and health of the population [6].

In Albania, this uneven distribution of the nursing workforce is evident, with major cities such as Tirana and Durres attracting the most health professionals due to more advanced infrastructure and better working conditions [7]. Tirana, for example, has 8430 nurses, while Durres has 1695. These numbers reflect a concentration that allows these areas to guarantee a more complete healthcare offer. In contrast, rural and remote regions, such as Kukës and Dibër, have much lower numbers of nursing staff, 854 and 1085 nurses, respectively, for populations of 71,498 and 106,380. The difference is particularly evident when considering the proportion of nurses in the population, with a lower ratio negatively impacting the ability of these regions to offer adequate care [8]. This disparity is attributable to a number of factors, including a lack of adequate healthcare infrastructure, less favorable working conditions, and limited professional development opportunities for nurses who choose to work in rural settings [9].

These inequalities reflect a global trend, where urban areas attract a higher density of health workers due to better living conditions and job opportunities, while rural areas remain disadvantaged [10]. Comparative studies show that countries such as Canada and Norway have adopted incentive policies to attract healthcare workers to remote areas, combining economic incentives, housing, and continuing education opportunities, which have led to a more balanced workforce redistribution [11]. These successful models could provide relevant examples for Albania, where the gap between urban and rural areas continues to widen. In many rural regions of Albania, the lack of infrastructure and the lack of technologies suitable for clinical work make the work environment difficult and limit the ability of nurses to provide comprehensive healthcare. This situation not only hampers patient well-being but also contributes to the exodus of nurses to cities, exacerbating the shortage in rural areas [12].

The Human Development Index (HDI) is a composite indicator developed by the United Nations to assess the socioeconomic progress of a region or country. This index combines three main dimensions: life expectancy at birth, the average and maximum levels of education attained, and gross national income per capita. A higher HDI score reflects better living conditions and broader access to essential services, including healthcare. In the context of our study, the HDI was used to identify and analyze regional disparities in the distribution of nursing personnel in Albania, providing a quantitative measure to examine how socioeconomic conditions influence access to healthcare resources.

The Human Development Index (HDI) provides further information on the disparities between the different regions of Albania. Areas with a higher HDI, such as Tirana (0.822) and Vlorë (0.804), are characterized by a higher density of nurses and a better supply of health services [7]. In contrast, less developed regions such as Dibër (0.756) and Kukës (0.754) suffer from significant shortages of health workers, a problem that contributes to exacerbating inequalities in access to care [13]. The gap in HDI between these regions indicates that citizens living in less developed areas face substantial challenges in accessing primary healthcare services, with direct implications on health outcomes. For example, the infant and maternal mortality rate tends to be higher in areas with a low density of nurses, given that the lack of continuous care prevents effective monitoring of health conditions and prevention of complications [14,15,16].

A further effect of inequalities in the distribution of nursing staff is the negative impact on the overall quality of healthcare services in rural areas. The shortage of nurses in remote regions increases the workload for existing staff, causing overload and a reduction in the quality of care [17]. Understaffing, coupled with long working hours, contributes to professional burnout and stress, which leads to lower efficiency and a higher likelihood of clinical errors [18]. Rural dwellers, lacking immediate access to adequate health facilities, are often forced to travel long distances to receive medical care, delaying treatment and increasing the risk of complications [19]. This situation is compounded by the difficulty of rural facilities to attract nurses, a problem that requires a targeted policy solution.

To quantify the level of inequality in the distribution of nurses in Albania, this study uses the Gini coefficient, a commonly applied tool to measure inequality in the distribution of resources within a population [20]. The Gini coefficient ranges from 0 to 1, where 0 represents perfect equality and 1 indicates extreme inequality. This measure provides a clear and numerical indication of regional inequalities, allowing the extent of the disparity in the distribution of nurses to be objectively assessed [21]. Using the Gini coefficient to analyze data on the nursing workforce in Albania allows you to identify areas that require immediate intervention and develop more targeted intervention strategies to improve access to health services across the country.

The data used in this study come from reliable sources, such as reports from the Albanian Ministry of Health and publications from the World Health Organization, which offer detailed information on the regional distribution of nurses, as well as the demographic and socioeconomic characteristics of the various regions of the country [7,20]. This information allows us to outline a complete picture of the health situation in Albania and to highlight the areas that need priority interventions to reduce inequalities in access to care. The use of up-to-date and reliable data is critical to ensure that conclusions are accurate and reflect current conditions, thus enabling policymakers to adopt evidence-based strategies to address the challenges of health workforce distribution.

The glaring disparity in the distribution of the nursing workforce in Albania requires an integrated and multidimensional policy approach. Albania could learn from other nations that have successfully implemented policies to incentivize and redistribute health resources in rural areas, such as Canada and Norway [11]. Such interventions could include economic incentives for nurses who choose to work in disadvantaged areas, investments in health infrastructure, and the promotion of continuing education programs to improve staff competence. Collaborating with international organizations, such as the World Health Organization and the European Union, could also provide technical and financial support to develop training programs and incentives for the health workforce, helping to reduce inequalities and ensure equitable access to care services for all Albanian citizens [20]. International cooperation could support Albania not only in terms of funding but also by providing guidelines and expertise to implement successful models adapted to the local context, increasing the effectiveness of interventions.

The implementation of long-term policies to improve working conditions and promote sustainable career growth in rural areas could significantly contribute to reducing inequalities. Albanian health authorities should develop specific career plans for nurses working in rural settings, offering them opportunities for professional advancement, specialization courses, and skills development programs [22]. In addition, creating psychological and professional support networks among nurses in these areas could help reduce feelings of isolation and improve the quality of working life, thus promoting a more stable and motivated nursing workforce.

Adopting a multidimensional approach based on economic incentives, investment in infrastructure, lifelong learning, and psychological support can not only help improve access to care in disadvantaged areas but also represent a key strategy to promote equity in the Albanian healthcare system. These interventions would not only improve the quality of life of patients in rural areas but would also have a positive impact on the working conditions of nurses, reducing turnover and improving the sustainability of the health system in the long term.

The aim of this study is to analyze regional disparities in the distribution of the nursing workforce in Albania, using the Gini coefficient and HDI as tools to identify inequities and propose evidence-based policy recommendations.

## 2. Materials and Methods

This descriptive–analytical study was conducted in 2024, with the aim of examining the distribution of the nursing workforce in Albania and identifying any inequalities between regions of the country. The descriptive–analytical design was chosen for its ability to provide a detailed overview of the current situation and, at the same time, to analyze the data collected in depth to draw meaningful conclusions about existing disparities.

### 2.1. Data Collection

Data on the number of nurses and the Albanian population were collected from authoritative and reliable sources. The main data sources include databases from the Albanian Ministry of Health and reports from the World Health Organization [7,20]. To ensure accuracy, the data were cross-verified with multiple reports and reviewed for consistency. All regions of Albania were included in the analysis, and no areas were excluded based on data availability or relevance. The data cover the geographical distribution of nurses in the various regions, the number of nurses per inhabitant, and the demographic characteristics of the health workforce, such as age, gender, and level of training.

For example, data show that Tirana, with a population of 925,268, has 8430 nurses, while the Kukës region has only 854 nurses for a population of 71,498. Other regions such as Dibër, with 1085 nurses per 106,380 inhabitants, also show a significant shortage of health personnel. In addition, the overall nursing workforce in Albania is mainly composed of women (80%), distributed among the age groups of 21–65 years. The use of these sources guarantees the validity and reliability of the data collected, being considered authoritative at international and national levels.

### 2.2. Calculation of the Gini Coefficient

The calculation of the Gini coefficient was performed to assess inequalities in the distribution of nurses across regions in Albania. The Gini coefficient was calculated using the following formula:G = 1 − ∑^i^ = _1n_ (Yi + Yi − 1)(Xi − Xi − 1)
where X represents the cumulative share of the population and Y represents the cumulative share of nurses. The analysis was conducted using SPSS version 28.0 and Microsoft Excel365 to ensure accurate calculations and replicability. This approach aligns with standard practices for regional inequality analysis.

### 2.3. Analysis Tools

To assess the distribution of nurses and identify regional disparities, the Gini coefficient, one of the most commonly used tools to measure inequality in the distribution of resources, was used. The Gini coefficient is a numerical indicator that varies between 0 and 1, where 0 indicates a perfectly equal distribution and 1 indicates a maximum inequality. The use of the Gini coefficient in this context is particularly useful for quantifying disparities in the distribution of nursing staff in relation to the population, comparing the levels of inequality between the various Albanian regions.

In addition to the Gini coefficient, the Lorenz curve was applied to graphically visualize the distribution of nurses with respect to the population. The Lorenz curve allows us to represent inequality, with the x-axis representing the cumulative population ordered by human development level (HDI), and the y-axis representing the cumulative distribution of nurses. If the curve approaches the perfect equality line, the distribution of nurses is more homogeneous; conversely, a farther curve represents greater inequality.

### 2.4. Classification of Regions

Albanian regions were ranked using the Human Development Index (HDI) provided by the United Nations. The HDI takes into account factors such as life expectancy, education level, and GDP per capita. Based on this index, the regions were divided into areas of high, medium, and low human development. Tirana (HDI 0.822) and Durres (HDI 0.804) are classified as high-development regions, with a higher concentration of nurses. In comparison, less developed regions such as Kukës (HDI 0.754) and Dibër (HDI 0.756) have chronic shortages of healthcare workers.

### 2.5. Analysis Procedure

Data analysis was conducted using advanced statistical software such as Microsoft Excel and SPSS, to ensure accurate data processing and calculation of the Gini coefficient and Lorenz curve. Each region was considered a separate unit of analysis, allowing us to examine variations within the country. The results were subsequently interpreted to draw conclusions about regional disparities in the distribution of nurses and to provide policy recommendations aimed at promoting a more equitable distribution of the nursing workforce.

## 3. Results

The analysis of the distribution of the nursing workforce in Albania shows significant inequalities between different regions. Table 1 shows the number of nurses, population, and Human Development Index (HDI) for each region, along with the ratio of nurses per 10,000 inhabitants. Regions with a higher HDI tend to have a higher concentration of nurses than less developed regions, but with some notable exceptions, as shown in Table 1.

Regions with a higher HDI, such as Tirana, have the highest number of nurses, with around 60 nurses per 10,000 inhabitants. In contrast, regions such as Kukës and Dibër, with lower HDI, have only 10 nurses per 10,000 inhabitants, highlighting a chronic shortage of health personnel. In these areas, the number of nurses is drastically lower than the demand for health services, further exacerbating the disparity in access to care.

However, analysis using the Gini coefficient, updated to 0.0228, shows a very low level of inequality in the distribution of the nursing workforce. While the overall Gini coefficient indicates a low level of inequality in the distribution of the nursing workforce, there are notable regional disparities. Urban areas such as Tirana benefit from a higher concentration of nursing personnel, whereas rural regions like Kukës and Dibër face significant shortages, highlighting the uneven distribution of resources within the country.

The Lorenz curve (Figure 1) graphically visualizes this equity, illustrating that the cumulative distribution of nurses is very close to the perfect equality line.

In addition to considering the distribution of nurses in individual regions, it is useful to analyze how the availability of nursing staff varies according to the Human Development Index (HDI) of different areas of the country. The regions were divided into three categories of HDI: low, medium, and high. Table 2 shows the total number of nurses, the total population, and the average ratio of nurses per 10,000 inhabitants for each category.

To better visualize the distribution of the nursing workforce by HDI, Lorenz curves were elaborated representing the distribution within groups of HDI and between groups of HDI compared to the total of regions.

The first graph (a) represents the Lorenz curve for the distribution of nurses within each HDI group, highlighting how the nursing workforce is distributed according to the socioeconomic context of the Albanian regions. The second graph (b) shows the Lorenz curve for the cumulative distribution of nurses between HDI groups and the total regions. Both graphs indicate a low level of inequality, as the curves approach the perfect equality line, showing a relatively equal distribution of the nursing workforce (Figure 2).

These regions, despite their relatively small population, have an above-average ratio of nurses per inhabitant. In particular, Gjirokastër and Kukës are rural regions with a good level of nursing coverage compared to their population, although they often lack advanced infrastructure (Table 3).

In contrast, urban regions such as Durres and Lezhë show a lower nurse-to-population ratio, despite a higher concentration of economic resources and population. This imbalance highlights the increasing pressure on health services in urban areas, where the population is larger, but the number of nurses per inhabitant is relatively low (Table 4).

### Effect of Socioeconomic Level and HDI on the Distribution of Nurses

HDI plays a crucial role in the distribution of the nursing workforce in Albania. Regions with higher HDI, such as Tirana (0.822) and Durres (0.804), attract more nurses due to advanced healthcare infrastructure and better working conditions. In contrast, regions with a lower HDI, such as Kukës (0.754) and Dibër (0.756), show a significant deficit in the nursing workforce, due to a lack of professional opportunities and economic resources.

The nursing workforce in Albania is predominantly composed of women (80%), but in rural regions, there is a higher proportion of male nurses, where nursing work is often more physically demanding. Regions with higher HDI, such as Tirana and Vlorë, demonstrate better nurse-to-population ratios, highlighting a positive correlation between socioeconomic development and workforce distribution. Conversely, areas with lower HDI, such as Kukës and Dibër, face significant challenges in ensuring equitable access to healthcare services, further exacerbating regional disparities.

The average age of nurses varies between urban and rural areas: in cities such as Tirana, many nurses are between the ages of 25 and 45, while in rural areas the workforce tends to be older, with a high proportion of nurses over 50. This imbalance suggests that rural regions could face a future shortage of nurses due to retirement, without adequate generational turnover.

The aging nursing workforce in rural regions like Kukës and Dibër, coupled with the limited presence of younger generations of nurses to replace them upon retirement, presents a critical challenge for the Albanian healthcare system. To address this, targeted policy measures are essential. Financial support to establish training programs in these regions could bolster the supply of new nurses. Furthermore, providing incentives for faculty to relocate to underserved areas would strengthen local nursing education. Investments in technology, such as advanced distance learning platforms, could also facilitate access to foundational and advanced training courses, especially in basic sciences. These initiatives would not only support workforce sustainability but also contribute to reducing regional disparities in healthcare provision.

The results of this study clearly highlight the need for targeted policies to further reduce inequalities in the distribution of the nursing workforce in Albania. The Gini coefficient of 0.0228 indicates a very low level of inequality, but it is still an invitation for health authorities to carefully monitor and intervene where necessary. To further improve distribution in low HDI areas, economic incentives, career growth opportunities, and continuing education programs could be implemented to encourage healthcare workers to stay in less developed regions.

In addition, there is a need to strengthen health infrastructure in rural areas in order to ensure a suitable working environment for nursing staff. This would not only improve the quality of care but also attract more qualified nurses to remote and less developed areas. With a combination of incentives and strategic policies, the Albanian government could hope to maintain the current low inequality and ensure increasingly equitable access to health services across the country.

## 4. Discussion

The results of this study highlight a significant inequality in the distribution of the nursing workforce in Albania, highlighting in particular the differences between urban and rural areas. The chronic shortage of nursing staff in rural areas results in limited access to care services for citizens and a lower quality of health services compared to urban areas. These inequalities undermine not only the effectiveness and efficiency of the Albanian healthcare system but also widen health disparities between different regions of the country, reflecting an equity issue that requires targeted interventions [23]. This situation is relevant not only for the well-being of the population but also for the sustainability of the health system in the long term, as an imbalanced distribution of the health workforce can contribute to systemic deficiencies that affect health outcomes at the national level [24,25].

Analysis of regional inequalities in the distribution of nurses shows that cities such as Tirana and Durres have significantly more nursing staff than less developed regions such as Kukës and Dibër. This trend is not unique to Albania but reflects a global trend in which urban areas and large cities attract more human resources due to better living conditions, more job opportunities, and better healthcare infrastructure [26]. Indeed, urban areas not only offer generally higher wages and better benefits, but also have easier access to continuing education programs, refresher courses, and a professional support network that is rarely available in rural regions [27]. As a result, nurses tend to prefer urban centers, leading to a shortage in rural areas, where the need for healthcare workers is often more pressing.

In contrast, rural and less developed areas of Albania suffer from a lack of adequate infrastructure and resources to support a sufficient number of nurses. In these regions, hospitals and clinics are often less equipped and have limited medical technology, which makes nursing work more complex and strenuous [28]. The lack of professional support and difficult working conditions, such as long shifts and high workloads, further disincentivize healthcare workers to work in these areas, exacerbating the already existing shortage [29]. Overload of work and lack of infrastructure not only affect the well-being of nurses, but also reduce the quality and safety of care for patients, creating a cycle of inefficiency and stress that further compromises the healthcare system.

This shortage of nursing staff in rural areas has a significant impact on the health outcomes of the population. Regions with a low nurse-to-population ratio show higher rates of infant and maternal mortality and worse outcomes in chronic disease management and responses to medical emergencies [30]. The scarcity of nurses makes it difficult to provide preventive care or regular monitoring of patients with chronic conditions, increasing the risk of avoidable complications and worsening overall health outcomes. In addition, the distance that patients have to travel to access well-resourced healthcare facilities is an additional obstacle, especially for the elderly, children, and people with reduced mobility.

Inequalities in the distribution of the nursing workforce also reflect a violation of the principle of equity in healthcare, which postulates that all citizens, regardless of their geographical location or socioeconomic status, have the right to equal access to health services. This inequality becomes even more critical when considering that rural areas are often the most vulnerable, with populations that may have lower health awareness and limited access to primary care [20]. A more equitable distribution of nurses among regions of the country could help reduce these disparities and improve the quality of life in underserved areas.

To address these inequalities, targeted policy interventions are needed. A first step could be to adopt economic incentives to attract nursing staff to rural and less developed areas. Proposed measures include increasing salaries, introducing bonuses for staying in difficult areas, and offering tax breaks for healthcare workers [31]. Such incentives could not only attract new nurses to rural regions, but also encourage existing ones to stay, reducing turnover and improving continuity of care. Countries such as Canada and Norway have successfully implemented incentive programs to attract health workers to remote regions, offering not only economic incentives but also the improvement of working conditions and the creation of supportive communities for health workers [32].

In addition to financial incentives, it is crucial to invest in continuing education programs for nurses working in rural areas. These programs should include refresher courses on medical technologies and advanced skills for the autonomous management of patients in low-resource settings [33]. Continuous training and professional support would ensure that nurses are adequately prepared to meet the unique challenges of rural areas and improve their ability to provide high-quality care even in difficult conditions. This type of training could also help reduce the sense of professional isolation that many nurses in these regions experience, improving job satisfaction and overall staff well-being.

Another crucial element in improving the distribution of nursing staff is investment in health infrastructure in rural areas. Without adequate facilities, even an increase in the number of nurses would not be enough to improve the quality of services [34]. Investments in well-equipped hospitals and clinics could not only improve access to care for patients but also make these regions more attractive to healthcare professionals. Modern and well-equipped facilities also provide a safer and less stressful working environment, helping to reduce turnover and improving the overall effectiveness of the healthcare system [20].

Finally, the implementation of long-term policies that improve working conditions and promote sustainable career growth in rural areas could further contribute to reducing inequalities. Health authorities should develop specific career plans for nurses working in rural settings, offering opportunities for professional advancement and specialization courses [9]. In addition, creating professional and psychological support networks for nurses in these regions could reduce isolation and increase the sense of belonging and morale of staff.

This study suggests that in order to effectively address inequalities in the distribution of the nursing workforce in Albania, a multidimensional strategy is needed that combines economic incentives, infrastructure investments, and continuing education programs. Collaborating with international organizations, such as the World Health Organization and the European Union, could also provide technical and financial support to develop effective incentive policies and programs [7,9,20]. An approach similar to that taken by other countries could offer benchmarks to improve equity and ensure that all Albanian citizens, regardless of their geographical location, have access to quality healthcare.

While the Results Section detailed the disparities in the distribution of the nursing workforce based on HDI and regional factors, this Discussion Section aims to contextualize these findings within broader socioeconomic and healthcare challenges. For instance, the observed disparities align with global trends where urban areas attract more resources, as highlighted in similar studies conducted in North Macedonia and Kosovo.

### 4.1. Study Limitations

Despite the significant results that emerged from this study, it is important to recognize some limitations that could affect the interpretation of the data and the generalizability of the conclusions. One of the main limitations concerns the availability and updating of data, as in some Albanian regions, especially in rural areas, data related to the nursing workforce and population may not be updated regularly or may lack accuracy, thus affecting the exact estimate of the nurse-to-population ratio and the Gini coefficient.

The study focuses predominantly on quantitative data regarding the distribution of nurses, neglecting a qualitative analysis that could provide a deeper understanding of the direct experiences of nurses and patients in underserved regions.

The inclusion of qualitative data, such as interviews or focus groups, would enrich the analysis, highlighting the operational challenges and motivations of nurses who choose to work in disadvantaged areas. Although the Human Development Index (HDI) was used as the main parameter to rank regions, other socioeconomic factors that could influence the distribution of nursing staff were not considered. Factors such as the level of regional health funding, local recruitment policies, and social infrastructure can have a significant impact on the ability to attract and retain health workers.

This study focuses exclusively on nurses, without considering the distribution of other crucial professional figures such as doctors, health assistants, and healthcare workers, whose inclusion could provide a more complete view of inequalities in the health sector in Albania.

In addition, the study provides a snapshot of the distribution of nurses at a specific point in time, without considering changes over time. A longitudinal analysis that examines the evolution of the distribution of the nursing workforce over the years could provide valuable insights into future trends and developments.

The study does not analyze in detail the effect of health migration, i.e., the transfer of nurses from rural to urban areas or from Albania to other countries, a factor that could significantly influence the availability of health workers in less developed areas and represents an important topic for future research.

These limitations suggest the need for further research that can fill these gaps and provide a more complete picture of the distribution of the healthcare workforce in Albania. However, despite these restrictions, the study makes a significant contribution to understanding inequalities in the distribution of the nursing workforce and suggests important implications for future policies.

### 4.2. Recommendations for Future Research

The results of this study highlighted significant inequalities in the distribution of the nursing workforce in Albania, suggesting different directions for future research that could broaden the understanding of this phenomenon and provide useful indications for more targeted policy interventions.

The study focused primarily on quantitative data regarding the distribution of nurses, but the inclusion of a qualitative analysis could provide a more complete picture. Qualitative surveys, through interviews or focus groups with nurses and health workers working in less developed areas, could provide detailed information on the challenges they face on a daily basis and the motivations that drive them to work or leave certain regions, thus enriching the analysis of geographical inequalities.

Internal migration (from rural areas to urban areas) and international migration (from one country to another country) are phenomena that could significantly affect the distribution of health personnel, and future studies could investigate the dynamics of this migration, understanding the motivations that push nurses to move and evaluating the impact of these movements on the Albanian health system.

A longitudinal analysis of the distribution of nurses over time could provide a clearer view of long-term trends. Monitoring the evolution of the nursing workforce over the years would identify variations due to health policies, socioeconomic changes, or migrations and could offer valuable data for long-term health planning.

Although this study focuses on nurses, other professional categories, such as doctors and health professionals, play a crucial role in the health system, and future research could examine the distribution of all health figures to gain a more comprehensive view of inequalities in the Albanian health sector, including the role of other crucial professions as well. It would also be useful to investigate more deeply the correlation between the uneven distribution of nurses and the health outcomes of the population.

Future studies could examine the direct impact of nursing staff shortages on health indicators, such as infant and maternal mortality rates, chronic disease management, and health emergency response, providing further arguments for more effective redistribution policies. The policies suggested in this study, such as economic incentives for health workers in rural areas, need to be evaluated to determine their effectiveness.

Future research could monitor the impact of such policies, once implemented, assessing whether these measures succeed in improving the distribution of nurses in less developed areas. Comparative studies between Albania and other countries with similar challenges in the distribution of health workforce could offer solutions adaptable to the Albanian context, exploring the best practices of other nations that have managed to rebalance the distribution of health resources could provide effective models for addressing disparities in Albania.

Another relevant area of research concerns demographic dynamics, such as age and gender, of the nursing workforce. In particular, it would be useful to explore the challenges related to the aging of nursing staff in rural areas and to develop strategies to foster generational turnover and incentivize young people to take up nursing in underserved regions.

Future research should focus on these areas to provide a deeper understanding of the dynamics that influence the distribution of the health workforce and to develop more effective strategies that aim to reduce inequalities and improve access to health services for all citizens.

### 4.3. Management Implications

The results of this study have important implications for healthcare management in Albania, highlighting the need for a strategic and targeted approach to address inequalities in the distribution of the nursing workforce and improve the overall effectiveness of the healthcare system.

One of the main priorities should be the development of effective policies for the redistribution of nursing staff in rural and less developed areas, incentivizing recruitment through competitive working conditions and access to opportunities for professional growth, together with retention measures such as professional support and benefits to avoid migration to urban areas or other countries.

In addition to global examples from countries like Canada and Norway, regional experiences in neighboring countries facing similar socioeconomic challenges provide relevant insights. For example, in North Macedonia, targeted financial incentives for healthcare workers in rural areas have improved retention rates, while Kosovo has implemented community-based training programs to address workforce shortages. Such localized strategies could serve as a blueprint for Albania.

Healthcare management should also develop workforce management plans based on long-term planning strategies, considering the aging workforce and the need for generational turnover, and supporting this planning with technological tools for continuous monitoring of the distribution of healthcare personnel.

It is essential to ensure that targeted investments in healthcare infrastructure are implemented in less developed areas, equipping hospitals and clinics with modern equipment and advanced technologies that effectively support the work of nurses, making these areas more attractive to healthcare workers.

Management should promote continuing education programs for nurses, with a focus on those working in rural settings and with limited resources, offering opportunities for specific training on emergency management, new technologies, and advanced clinical skills to increase the capacity of staff to respond to challenges in difficult contexts.

It is crucial to adopt policies that provide economic incentives to attract staff to disadvantaged areas, such as increased salaries, housing, or transport costs, as part of a broader strategy to improve working conditions in underserved regions.

In addition, management should implement psychological support programs for staff, especially those working in high-pressure settings, for stress management and burnout prevention, promoting a sustainable working environment and improving the quality of care.

The adoption of policies for the improvement of working conditions in rural areas, with manageable workloads, sustainable shifts, and adequate supervision, can improve the well-being of staff and promote an organizational culture based on the enhancement of nursing work.

It is crucial for management to work with educational institutions and government authorities to develop curricula that prepare new nurses to work in challenging settings, create internship opportunities in rural areas, and work with the government to implement reforms that support an equitable distribution of human resources in the healthcare system.

The proposed interventions in this study are supported by evidence from international literature. For instance, studies have demonstrated the effectiveness of financial incentives, professional development opportunities, and technology-driven training programs in addressing workforce imbalances in resource-constrained settings. Implementing these approaches in Albania would require careful adaptation to the local context, including addressing barriers such as funding limitations and policy challenges.

## 5. Conclusions

The uneven distribution of nurses in Albania is one of the main causes of serious inequalities in access to health services. Regions with a low Human Development Index, particularly rural areas such as Kukës and Dibër, suffer from a chronic shortage of nursing staff. This disparity translates into lower quality of healthcare and greater difficulties in accessing essential services, further exacerbating already existing socioeconomic inequalities.

The results of this study highlight the urgency of targeted interventions by the Albanian government and health institutions to rebalance the distribution of nursing resources. It is crucial to adopt an integrated strategy that combines economic incentives, continuing education programs, and improved health infrastructure in less developed areas.

Economic incentives are a crucial first step in attracting and retaining nurses in rural regions. Higher salaries, bonuses for staying in difficult areas, and tax breaks could incentivize health workers to work in areas that currently suffer from a chronic shortage of human resources. However, economic incentives alone are not enough.

Continuing education is essential to ensure that nurses are adequately prepared to deal with the specific challenges of low-resource settings. Targeted refresher and specialization programs in emergency areas, chronic disease management, and modern medical technologies are indispensable to improve the quality of care provided in less developed regions.

Improving healthcare infrastructure is another key pillar. Without properly equipped hospitals and clinics, nursing staff will not have the resources they need to do their jobs effectively. Improving healthcare facilities in these areas would not only make nursing work more sustainable in the long term, but it would also help provide quality care to patients, reducing the need for travel to other regions to receive specialized healthcare.

It is crucial to adopt a long-term approach that includes professional development policies and support for nursing staff. Providing opportunities for career advancement and promoting the growth of nurses’ skills, especially in disadvantaged areas, will help ensure a fairer and more sustainable healthcare system.

Future research should focus on exploring strategies to retain nursing staff, particularly in underserved areas, and evaluating the impact of migration on workforce distribution. Additionally, addressing challenges such as funding limitations and policy implementation barriers will be crucial for ensuring the success of proposed interventions. By addressing these gaps, future studies can provide a more robust foundation for developing equitable and sustainable healthcare workforce policies.

Only through a combination of policy interventions, addressing both the immediate and structural causes of inequalities, will it be possible to effectively reduce regional disparities and ensure universal and equitable access to health services in Albania. It is a complex challenge, but with targeted policies and constant commitment, Albania can build a health system that guarantees quality care to all citizens, regardless of their geographical location.

## Figures and Tables

**Figure 1 nursrep-15-00030-f001:**
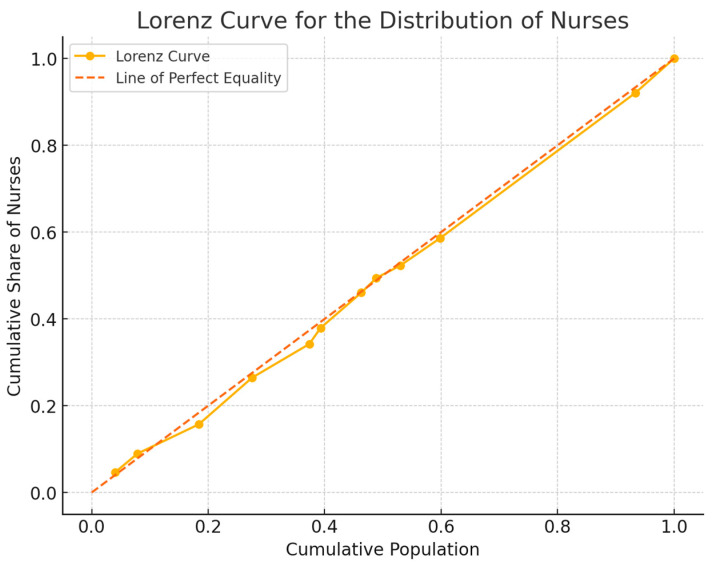
Lorenz curve for nurse workforce distribution in Albania.

**Figure 2 nursrep-15-00030-f002:**
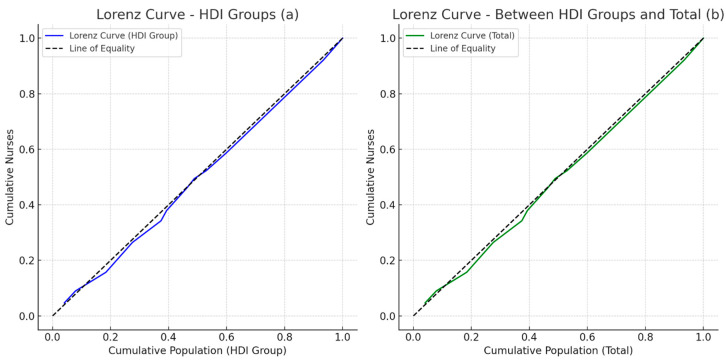
Lorenz curves for nurse distribution in Albania based on HDI.

**Table 1 nursrep-15-00030-t001:** Distribution of nurses by region in Albania: number, population, and Human Development Index (HDI).

Region	Number of Nurses	Population	HDI	Nurses per 10,000 Inhabitants
BERAT	1177	111.31	0.783	10.56
DIBER	1085	106.380	0.756	10.20
DURRES	1695	289.797	0.803	5.85
ELBASAN	2708	252.719	0.785	10.72
FIER	1959	271.672	0.769	7.21
GJIROKASTER	932	53.314	0.797	17.49
KORCE	2047	192.925	0.792	10.61
KUKES	854	71.498	0.754	11.94
LEZHE	727	114.181	0.771	6.37
SHKODER	1593	189.164	0.785	8.42
TIRANE	8430	925.268	0.822	9.11
VLORE	1997	183.436	0.804	10.89

**Table 2 nursrep-15-00030-t002:** Distribution of the nursing workforce by HDI categories.

HDI Category	Total Nurses	Total Population	Average Nurses per 10,000 People
Low HDI	2766	290.809	9.09
Medium HDI	7399	942.259	8.78
High HDI	10.447	1720.949	10.73

**Table 3 nursrep-15-00030-t003:** The three regions with the highest ratio of nurses per 10,000 inhabitants.

Region	Number of Nurses	Population	Nurses per 10,000 Inhabitants
GJIROKASTER	932	53.314	17.49
KUKES	854	71.498	11.94
ELBASAN	2708	252.719	10.72

**Table 4 nursrep-15-00030-t004:** The three regions with the lowest ratio of nurses per 10,000 inhabitants.

Region	Number of Nurses	Population	Nurses per 10,000 Inhabitants
LEZHE	727	114.181	6.37
DURRES	1695	289.797	5.85
FIER	1959	271.672	7.21

## Data Availability

The data presented in this study are available on request from the corresponding author.

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
