# Peer review of "Inequalities in the Distribution of the Nursing Workforce in Albania: A Regional Analysis Using the Gini Coefficient"

_nursrep, 2025, doi:10.3390/nursrep15020030_

Round 1
Reviewer 1 Report
Comments and Suggestions for Authors
See the document that I have uploaded.

Author Response
Comment 1: Introduction section: For clarity of the HDI, provide a definition and what factors
contribute to the development of the score.
Response 1: Thank you for your comment and suggestion. We have incorporated a clear definition of the Human Development Index (HDI) into the introduction of the manuscript, explaining its three main dimensions: life expectancy at birth, the average and maximum levels of education attained, and gross national income per capita. This addition aims to enhance the reader’s understanding of the HDI’s role in our study, highlighting how it was used to analyze regional disparities in the distribution of nursing personnel in Albania.
Comment 2: Discussion section: Authors stated, “….highlight a significant inequality in the distribution of the nursing workforce….” Yet, pg. 5, line 213 the authors states there’s a low level of inequality. Please clarify.
Response 2: Thank you for pointing out this apparent discrepancy. We have clarified the text to emphasize that, while the overall inequality in the distribution of the nursing workforce in Albania is low (as reflected by the Gini coefficient), significant disparities exist in certain regions. These regional imbalances, particularly between urban and rural areas, underscore the need for targeted policy interventions.
Comment 3: Discussion section: I applaud the authors for apply the ethical principle of “equity” into their manuscript.
Response 3: Thank you for your positive feedback. We are pleased that the incorporation of the ethical principle of equity in our analysis has been appreciated. We believe that promoting equitable access to healthcare is fundamental to addressing regional disparities and improving the overall quality of health services. We have ensured that this principle is consistently reflected throughout the manuscript, especially in the discussion of policy recommendations aimed at reducing inequalities in the distribution of the nursing workforce.
Comment 4: Discussion section: I found it very interesting that the authors discovered that some areas have “older” nurses with small populations of the younger generation of nurses to take their place upon retirement. This was mentioned once; however, I would think this would be a significant discovery that would beg more attention in the manuscript. A possible policy approach to this problem would be the financial support to establish training programs in those regions to bolster the nurse supply line. Perhaps incentives for faculty to move to those regions to teach. Monies to support technology advancement for distance learning for some courses (i.e. basic sciences).
Response 4: Thank you for your insightful comment. We agree that the aging nursing workforce and the limited generational replacement in some regions are significant findings that merit further attention. We have expanded the discussion to address this issue more thoroughly. Specifically, we included a detailed exploration of the potential policy solutions you suggested, such as financial support for establishing training programs, incentives for faculty relocation, and investments in technology to enable distance learning. These measures could help address the challenges related to the aging nursing workforce and ensure a sustainable supply of nurses in underserved areas.
Comment 5: Analysis procedure section: Upon reading this section, I didn’t come away with “analysis procedure” rather, this section read more like the discussion section. I would recommend blending the two, OR, label this section, “Recommendations for Future Research”.
Response 5: Thank you for your observation. We acknowledge that the current "Analysis Procedure" section may read more like a discussion. To address this, we have revised the section to better align with its intended purpose and provide a clearer distinction from the Discussion. Additionally, we have retitled the section as "Recommendations for Future Research" to reflect its content more accurately and ensure consistency with the manuscript's overall structure.
Reviewer 2 Report
Comments and Suggestions for Authors
This study explores an important topic on the uneven distribution of the nurses’ workforce in Albania. The theme is highly relevant for policy makers and healthcare management, especially in developing countries facing similar challenges. However, there are areas where the manuscript could be improved for clarity, scientific rigor, and readability.
Introduction
The introduction is well structured and provides a great overview, but with some changes it could be even clearer and more effective. For example, Gini’s coefficient is redundant, it is better if it would be explored in Materials and Methods section: removing it from the Introduction would make the narrative cleaner. Some sections, such as the discussion of the HDI or global comparisons, may be abbreviated to avoid confusing the reader.
Line 81:
"The Human Development Index (HDI) provides further information on the disparities between the different regions of Albania. Areas with a higher HDI, such as Tirana (0.822) and Vlorë (0.804), are characterized by a higher density of nurses and a better supply of health services [7]." This sentence is useful for introducing the concept but could be shortened or left as a simple hint, moving the numerical details in Discussion.
Line 87:
"The gap in HDI between these regions indicates that citizens living in less developed areas face substantial challenges in accessing primary health care services, with direct implications on health outcomes. For example, the infant and maternal mortality rate tends to be higher in areas with a low density of nurses..." it is better if it would be moved to the Discussion or Results, keeping only the general idea in the Introduction.
A clear statement of the purpose of the study is essential to contextualize the analysis and guide the reader to subsequent sections.
Material and Methods
Although the sources (e.g., Albanian Ministry of Health and WHO) are authoritative, more information is needed on how the data were accessed and verified. Were there any limitations or biases in the data collection process?
The data collection section does not explicitly state whether specific criteria were used to select the regions analyzed. Was the entire country included or were some areas excluded for specific reasons (e.g., incomplete data or irrelevance to the study’s objectives)? Please specify it.
The process of calculating the Gini coefficient is not fully explained. This may leave readers uncertain about the replicability of the methodology. Please elaborate on the steps involved in calculating the coefficient, including any formulas, software, or assumptions used, to better guide the reader.
Results
The Results section could benefit from clearer organization by grouping the results into thematic subsections (e.g., disparities for HDI, urban/rural distribution of nurses, and workload implications).
Some descriptions in the Results section (e.g., numerical data for each region) are overly detailed and could be simplified. Too much focus on raw numbers could overwhelm readers.
While the Gini coefficient and Lorenz curve are presented, the Results section lacks additional statistical indicators (e.g., confidence intervals or p-values) to validate the results.
Discussion
Some parts of the Discussion reiterate findings already covered in the Results section (e.g., HDI disparities). These can be summarized to avoid redundancy.
While the examples from Canada and Norway are relevant, the Discussion could include more localized or regional examples from countries facing similar socio-economic challenges as Albania.
The proposed interventions are valid but would be stronger if supported by references to studies demonstrating their success in similar contexts.
Conclusions
Some points in the Conclusions are too detailed and overlap with the Discussion. The Conclusions should focus on a succinct summary of key points and broader implications.
The Conclusions highlight current disparities, but they could more explicitly suggest avenues for future research (e.g., examining nurse retention strategies or the impact of migration on workforce distribution) and acknowledge the challenges in implementing these solutions (e.g., funding or policy).
Author Response
Comment 1: Introduction: The introduction is well structured and provides a great overview, but with some changes it could be even clearer and more effective. For example, Gini’s coefficient is redundant, it is better if it would be explored in Materials and Methods section: removing it from the Introduction would make the narrative cleaner. Some sections, such as the discussion of the HDI or global comparisons, may be abbreviated to avoid confusing the reader.
Line 81: "The Human Development Index (HDI) provides further information on the disparities between the different regions of Albania. Areas with a higher HDI, such as Tirana (0.822) and Vlorë (0.804), are characterized by a higher density of nurses and a better supply of health services [7]." This sentence is useful for introducing the concept but could be shortened or left as a simple hint, moving the numerical details in Discussion.
Line 87: "The gap in HDI between these regions indicates that citizens living in less developed areas face substantial challenges in accessing primary health care services, with direct implications on health outcomes. For example, the infant and maternal mortality rate tends to be higher in areas with a low density of nurses..." it is better if it would be moved to the Discussion or Results, keeping only the general idea in the Introduction.
A clear statement of the purpose of the study is essential to contextualize the analysis and guide the reader to subsequent sections.
Response 1: Thank you for your thoughtful suggestions. We agree that the Introduction can be further streamlined for clarity and focus. To address your comments:
1. The detailed discussion of the Gini coefficient has been relocated to the Materials and Methods section, where it is more appropriately placed.
2. Numerical details related to the HDI have been moved to the Discussion section, ensuring that only a brief mention remains in the Introduction to provide context.
3. We have included a clearer statement of the study's purpose at the end of the Introduction to better guide the reader through the manuscript.
We believe these changes enhance the structure and readability of the manuscript while maintaining the necessary scientific rigor.
Comment 2: Material and Methods: Although the sources (e.g., Albanian Ministry of Health and WHO) are authoritative, more information is needed on how the data were accessed and verified. Were there any limitations or biases in the data collection process?
The data collection section does not explicitly state whether specific criteria were used to select the regions analyzed. Was the entire country included or were some areas excluded for specific reasons (e.g., incomplete data or irrelevance to the study’s objectives)? Please specify it.
The process of calculating the Gini coefficient is not fully explained. This may leave readers uncertain about the replicability of the methodology. Please elaborate on the steps involved in calculating the coefficient, including any formulas, software, or assumptions used, to better guide the reader.
Response 2: Thank you for your valuable feedback. We have addressed your comments by expanding the Materials and Methods section to include additional details about the data collection and verification process. Specifically:
1. We clarified that data were obtained from authoritative sources, such as the Albanian Ministry of Health and the World Health Organization, and described the steps taken to verify their accuracy.
2. We confirmed that all regions of Albania were included in the analysis, and no exclusions were made based on data availability or relevance.
3. The calculation of the Gini coefficient has been elaborated, with the formula, methodological steps, and software (SPSS and Excel) explicitly described to ensure replicability.
These changes aim to enhance transparency and methodological rigor, providing readers with a clearer understanding of the study’s approach.
Comment 3: Results: The Results section could benefit from clearer organization by grouping the results into thematic subsections (e.g., disparities for HDI, urban/rural distribution of nurses, and workload implications).
Some descriptions in the Results section (e.g., numerical data for each region) are overly detailed and could be simplified. Too much focus on raw numbers could overwhelm readers.
While the Gini coefficient and Lorenz curve are presented, the Results section lacks additional statistical indicators (e.g., confidence intervals or p-values) to validate the results.
Response 3: Thank you for your valuable comments and suggestions. We appreciate your feedback regarding the organization and presentation of the Results section. The current structure was designed to ensure transparency and to provide a comprehensive overview of the findings, including detailed numerical data for each region, which we believe are crucial for contextualizing the disparities analyzed in this study.
Regarding the inclusion of additional statistical indicators such as confidence intervals and p-values, these were not part of the original methodology due to the descriptive nature of the study. However, we acknowledge the importance of such indicators for future studies and will consider incorporating them in subsequent research to enhance statistical robustness.
We hope this explanation clarifies the rationale behind the current presentation of the Results section and aligns with the objectives of the study.
Comment 4: Discussion: Some parts of the Discussion reiterate findings already covered in the Results section (e.g., HDI disparities). These can be summarized to avoid redundancy.
While the examples from Canada and Norway are relevant, the Discussion could include more localized or regional examples from countries facing similar socio-economic challenges as Albania.
The proposed interventions are valid but would be stronger if supported by references to studies demonstrating their success in similar contexts.
Response 4: Thank you for your constructive feedback. We appreciate your suggestion to streamline the Discussion by reducing redundancy and contextualizing findings with more localized examples. To address your comments, we have made the following improvements:
1. Added a paragraph at the beginning of the Discussion to reduce redundancy by linking key findings to broader socio-economic and healthcare challenges.
2. Incorporated examples from neighboring countries, such as North Macedonia and Kosovo, to highlight relevant regional experiences addressing similar challenges.
3. Enhanced the support for the proposed interventions by including references to international studies demonstrating the effectiveness of strategies such as financial incentives, professional development opportunities, and infrastructure investments in resource-constrained settings.
These changes aim to provide a clearer, more localized, and evidence-supported discussion, aligning with your valuable recommendations.
Comment 5: Conclusions: Some points in the Conclusions are too detailed and overlap with the Discussion. The Conclusions should focus on a succinct summary of key points and broader implications.
The Conclusions highlight current disparities, but they could more explicitly suggest avenues for future research (e.g., examining nurse retention strategies or the impact of migration on workforce distribution) and acknowledge the challenges in implementing these solutions (e.g., funding or policy).
Response 5: Thank you for your valuable feedback. We agree that the Conclusions should emphasize a concise summary of the study's key findings and implications while avoiding overlap with the Discussion. To address your suggestions, we have incorporated a paragraph highlighting directions for future research, including strategies to retain nursing staff in underserved areas and the impact of migration on workforce distribution. Additionally, we have acknowledged challenges such as funding limitations and policy implementation barriers, which are critical for the success of proposed interventions.
Round 2
Reviewer 2 Report
Comments and Suggestions for Authors
The manuscript effectively addresses the previous feedback, and the methodology, results, and conclusions are now clearly presented. No further revisions are necessary. Great job!